# An Alpha/Beta Radiation Mapping Method Using Simultaneous Localization and Mapping for Nuclear Power Plants

**Xin Liu [1], Lan Cheng [1,\*], Yapeng Yang [2], Gaowei Yan [1], Xinying Xu [1] and Zhe Zhang [1]**

1   College of Electrical and Power Engineering, Taiyuan University of Technology, Taiyuan 030024, China
2   China Institute for Radiation Protection, Taiyuan 030024, China
*   Correspondence: chenglan@tyut.edu.cn

**Abstract:** Nuclear safety has always been a focal point in the field of nuclear applications. Mobile robots carrying specific sensors for nuclear-radiation detection have become an alternative to manual detection. This work presents an autonomous $\alpha/\beta$ radiation mapping framework, using a mobile robot carrying a light detection and ranging (LiDAR) and a nuclear-radiation-detection sensor. The method employs simultaneous localization and mapping (SLAM) techniques and radiation-detection sensors. Cartographer is adopted as a demonstration example to map the unknown environment. Radiation data are obtained through the radiation detection sensor and projected onto the environment map after coordinate alignment. The color-coded radiation map is shown on the environment map according to the dose rate. The simulation and real-environment experiments in a robot-operating system (ROS) validate the effectiveness of the proposed method in different radiation scenarios for both indoor and outdoor environments.

**Keywords:** SLAM; map fusion; nuclear radiation; surface contamination





## 1. Introduction

The threat of nuclear leakage is growing as nuclear energy becomes more widely used in the new energy-generation field. Quantification of radiation dose in and around a nuclear facility (radiation mapping) is crucial. Thanks to advances in robotics and the increasing concerns about human and environmental safety [1], using robots carrying radiation-detection sensors has become a promising alternative to manual detection, as it avoids human exposure to radiation.

Different radiation-detection methods are used according to the types of radiation. There are two common types of radiation in nuclear applications: charged particles, such as $\alpha$-rays, $\beta$-rays, and protons; and uncharged particles, such as X-rays, $\gamma$-rays, and neutrons. Currently, most of the detection of nuclear radiation is for X-rays and $\gamma$-rays [2], while there seems to be relatively little research on the detection of $\alpha/\beta$ radiation. Although $\alpha/\beta$ radiation is relatively easier to shield, both internal exposure from $\alpha$-rays and external exposure from $\beta$-rays can cause substantial damage to the human body. In addition, the accurate detection of $\alpha/\beta$ radiation allows the location of the radiation source to be identified, thus avoiding radiation interference with the micro-components in the integrated circuit board. Detection for $\alpha/\beta$ radiation can be divided into three categories: direct detection, indirect detection, and scan detection [2].

Both direct and indirect measurement methods, mainly operated by humans in the potentially contaminated area, are unable to measure total regional radiation levels, and the accuracy of the measurements is hardly guaranteed. On the other hand, the scan-detection method, which achieves autonomous radiation detection by fixing a radiation detector to a mobile robot, has attracted attention in recent years [3]. Scan detection allows for better radiation detection due to the robot's autonomous exploration capabilities. However,

detecting the radiation intensity and its location and then projecting the results on an environment map for better man–machine interaction is still an open problem. SLAM is an effective technique to solve this problem.

In the absence of a priori environmental information, SLAM uses a robot carrying specific sensors to model the environment during motion, while estimating its motion [4]. SLAM has become one of the essential technologies for the autonomous localization and navigation of unmanned systems since it does not rely on satellite signals but only uses sensor information to achieve localization and navigation. An intuitive idea to achieve autonomous nuclear-radiation detection is to equip the robot with a radiation detector and build a map of the environment by using SLAM methods. The radiation-detection results are then aligned and fused with the scene map. Autonomous nuclear-radiation detection consists of two steps: scene map building and the fusion of radiation detection results with the scene map.

Scene mapping uses SLAM to represent information about the scene. Maps have different forms depending on their characteristics and applications, such as sparse-feature maps, point cloud maps, topological maps, and grid maps [5–8]. In order to enable the robot to navigate autonomously through the radiation environment, as well as follow a specified path to detect the radiation results, the prerequisite is to build a grid map of the radiation scene. The grid-map-building method was first proposed by Elfes [9]. The idea is to divide the entire working environment into homogeneous grids and propose the use of probabilities to indicate the likelihood of obstacles in each grid. The grid map plays an important role in numerous robotic applications such as localization [10], exploration [11], and mapping performance benchmarking [12]. The Gmapping algorithm is an open-source SLAM algorithm based on a filtering SLAM framework, which is also commonly used for grid map building [13]. The Gmapping algorithm is widely used to implement SLAM in robots with LiDAR and overcomes the disadvantages of particle dissipation of the RBPF algorithm [14]. Today, the most famous open-source SLAM algorithm is Cartographer, which was developed by Google [15]; it combines LiDAR and inertial measurement unit (IMU) data to generate a real-time two-dimensional grid map.

The projection of the radiation results on the scene map is essentially a multi-source-information-fusion problem. The basic principle is to make full use of multi-sensor observation information. According to certain criteria, the complementary information from multiple sensors is combined in space or time to obtain consistent information. Asharif et al. proposed the use of Bayes's method and De Morgan's rule in a grid map to accomplish the fusion of different sensor data [16]. Besada-Portas et al. proposed a delay estimation TDE algorithm for mobile chassis environment-aware sensor fusion [17]. This algorithm completes the data synchronization of all types of sensors according to the timestamp, thus ensuring the accuracy of all types of sensor inputs and the fused model. Krystian Chachuła et al. proposed an enhanced algorithm for multi-sensor data fusion that is used for the detection of pollutants in wastewater [18]. Kai et al. combined scene-data fusion (SDF) with localization and mapping (LAMP) to propose a method for imaging three-dimensional $\gamma$ nuclear-radiation sources [19]. Manish K. Sharma et al. proposed a statistic-based grid-refinement method for backtracking the position of a $\gamma$-rays source in a three-dimensional domain in real time [20].

With regard to the location of different types of nuclear radioactive sources, researchers have been focusing on estimating the location of radioactive sources from radiation dose rates measured by nuclear-radiation detectors on mobile robots. Typical methods include the least-squares methods, maximum likelihood estimation methods, geometric methods, and Bayesian estimation algorithms [21–24]. These methods are good choices when it comes to X-rays and $\gamma$-rays since these rays can travel to a much farther distance compared to $\alpha/\beta$ radiation, and the detector has to determine the source location according to the radiation data.

The application of mobile robots to monitor radioactive areas has increased significantly since the Fukushima Daiichi nuclear power plant leak in 2011. Currently, most of the

mobile robots that have been designed for radiological features of the Fukushima Daiichi nuclear power plant are applied for γ-rays detection. For instance, the JAEA-3 [25] and Quince [26] robots have been deployed for γ-rays monitoring at the Fukushima nuclear facility. The RICA robot was developed by the French Alternative Energy and Atomic Energy Commission and Cyberia France, which has also been deployed to operate nuclear facilities for γ-ray detection [27]. In addition, the Georgia Institute of Technology developed a suite of robots for detecting and locating embedded γ-rays and neutron sources [28]. However, research on mobile robots for α/β radiation detection is not yet sufficient. Though the CARMA robot developed by the University of Manchester in collaboration with Sellafield Site is capable of α/β radiation detection [29], the CARMA platform is only applicable to a small range of scenarios and cannot meet the needs of a large range of environments such as nuclear plants.

As α/β radiation dies out when it is even a few centimeters away from the radiation source, we can take the position where the radiation is detected as the location of the radiation source. So, different from the aforementioned work, this paper presents a mobile robot solution that can be used for autonomous detection of α/β radiation. In this study, the surface-contamination detector was equipped with a Turtlebot2 robot for α/β radiation monitoring. In particular, the surface-contamination detector is a self-developed radiation detector that can be used for surface-contamination monitoring [30]. In addition, the methodological framework proposed in this paper is also applicable to the X-rays/γ-rays environment if the appropriate sensor is utilized, meaning that, once the robot pose is obtained, we can always locate the source of radiation since we know the relative coordinate between the robot and the radiation detector when the hardware is designed.

This paper is organized as follows. Section 2 describes the hardware materials and method proposed in this paper. Section 3 presents the simulation and experiment results. The analysis of the results is also conducted in this section. Section 4 discusses the experimental details of this paper and its contributions. Section 5 concludes the paper.

The abbreviations used in this paper are listed in Table 1.

**Table 1.** The abbreviations used in this paper.

| Abbreviations | Full Name |
| --- | --- |
| LiDAR | Light detection and ranging |
| SLAM | Simultaneous localization and mapping |
| IMU | Inertial measurement unit |
| LAMP | Scene-data fusion |
| SPA | Sparse pose adjustment |
| BBS | Branch-and-bound scan matching |
| TCP/IP | Transmission control protocol/internet protocol |
| RMSE | Root mean square error |
| TF | Transform frame |
| ROS | Robot operating system |
| STD | Standard deviation |
| SSE | Sum of squares error |
| APE | Absolute trajectory error |

## 2. Materials and Methods

### 2.1. Hardware Materials

#### 2.1.1. Robotics Platform

This study used Turtlebot2 as a vehicle for radiation detection and map building. Turtlebot2 is a robotic platform manufactured by Willow Garage, California, USA. Turtlebot2 is shown in Figure 1. It includes NVIDIA JETSON TX2, KOBUKI chassis.RPLiDAR A2 LiDAR, etc.

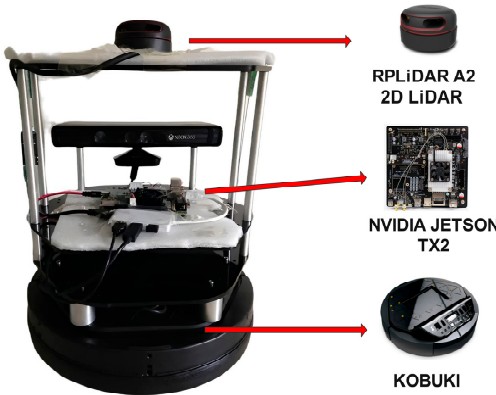

**Figure 1.** Robotics platform.

The Turtlebot2 is driven by KOBUKI and is equipped with a three-axis digital gyroscope, collision sensor, cliff sensor, and wheel encoder. KOBUKI is a robotic chassis from Yujin Robot that is widely used as a robot research platform. The odometry information is provided by the wheel encoder. The Turtlebot2 is equipped with the NVIDIA JETSON TX2, which is an embedded system developer module. The mobile robot platform is also equipped with LiDAR and a surface-contamination monitor. The LiDAR enables real-time distance and angle information to be measured between the mobile robot and its surrounding obstacles. This information is mainly used by the mobile robot for the creation of a two-dimensional grid map of the searching area and positioning and obstacle avoidance when it comes to autonomous navigation. The LiDAR used in this study is RPLiDAR A2, which has a maximum effective measuring distance of 40 m and an effective scanning angle of 360 degrees. The surface-contamination monitor used in this system was developed by the China Institute for Radiation Protection [30]. More detail is given in the next subsection.

### 2.1.2. Surface Contamination Monitor

The surface-contamination monitor used for radiation mapping is shown in Figure 2, and the main parameters are given in Table 2.

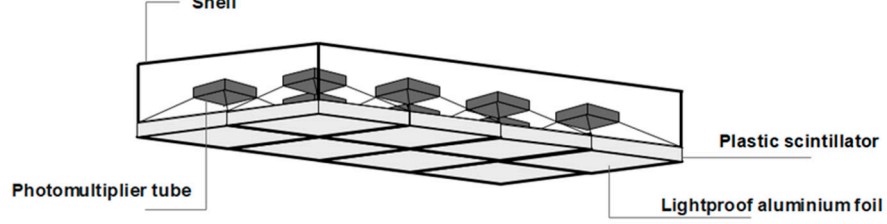

**Figure 2.** Surface-contamination monitor's construction diagram.

**Table 2.** Main parameters of the surface-contamination monitor.

| Performance | Parameters |
|---|---|
| Measured energy range | a > 2.5 MeV, β > 100 keV |
| Detection efficiency | a: $\geq$ 25%(Am-241); β: $\geq$ 30%(Sr-90/Y-90) |
| Effective area | $10 \times 17 \text{cm}^2 \times 8$ |
| Operating efficiency | $\geq$3 m$^2$/min |

Figure 2 shows the construction of the surface-contamination monitor, which consists of a shell, photomultiplier tube, plastic, scintillator, and light-proof aluminum foil. The surface-contamination monitor operates at a distance of 0.5–1 cm from the ground, and the measurement mode is a two-channel counter, which measures $\alpha/\beta$ radiation dose values separately.

The detection principle is that, when $\alpha/\beta$ particles enter the sensitive zone of the composite scintillator, the particles interact with the respective sensitive detection media in the composite scintillator to produce light pulses of different amplitudes. The light-pulse signal is transformed into an electrical-pulse signal through the photomultiplier tube. After preprocessing by the signal-processing unit, the count rates of the $\alpha$ and $\beta$ signals are recorded by the microcontroller system, and the detection results are then output.

In the radiation-detection process, when the $\alpha/\beta$ radiation-contamination area is smaller than the detector window area, the surface-contamination monitor is used to detect the contamination plane directly. The surface emissivity of the contaminated area and the surface activity of the contaminated area are calculated according to Equations (1) and (2):

$$q = \frac{N_i - N_b}{R_i} \tag{1}$$

$$A = Q \times \frac{N_i - N_b}{R_i \times S} \tag{2}$$

where $q$ is the surface emissivity, with unit $(s \times 2\pi \mathrm{sr})^{-1}$; $N_i$ is the average of the instrument's dose rate, with unit $\mathrm{s}^{-1}$; $N_b$ is the average of the instrument's background measurements, with unit $\mathrm{s}^{-1}$; $R_i$ is the surface emissivity response of the corresponding magnitude of the nuclide, with unit Bq; $A$ is the surface activity, with unit $\mathrm{Bq/cm}^2$; $Q$ is the ratio of the standard surface activity to surface emissivity; and $S$ is the area of the detection window of the surface contamination monitor, with unit $\mathrm{cm}^2$.

When the $\alpha/\beta$ radiation-contamination area is larger than the detector-window area, the effect of uneven contamination areas on the measurement results must be taken into account. The presence of inhomogeneous contaminated areas is determined by moving the detector to measure different contaminated areas. If the contaminated area is homogeneous, radiation detection is carried out directly by using a surface-contamination monitor. The surface emissivity of the contaminated area and the surface activity of the contaminated area are calculated according to Equations (3) and (4), respectively:

$$q = \frac{(N_i - N_b) \times S_1}{R_i \times S} \tag{3}$$

$$A = Q \times \frac{(N_i - N_b) \times S_1}{R_i \times S^2} \tag{4}$$

where $S_1$ is the area of the contaminated area.

When the contaminated area is not homogeneous, the multi-point measurement method can be applied. After selecting different locations in the contaminated area for measurement and calculating the surface emissivity and surface activity at each point in the contaminated area according to Equations (3) and (4), the average of the results at each point is considered as the measurement of the contaminated area.

### 2.2. LiDAR SLAM

In this section, some common LiDAR SLAM algorithms are introduced. The basic process of LiDAR SLAM involves a mobile robot starting from any area in an unknown environment and building a map of the environment, while obtaining environmental information based on the LiDAR it carries. The SLAM problem can be formulated as follows:

$$p(x_{1:k}, m | z_{1:k}, u_{k-1}) \tag{5}$$

where $z_{1:k}$, $u_{k-1}$ represents the measurement from LiDAR and the control input, $x_{1:k}$ represents the pose of the robot from time 1 to time $k$, and $m$ is the map built till time $k$.

The SLAM problem is an estimation problem. Based on the estimation theory employed, there are two categories of SLAM approaches: filter-based approaches and nonlinear optimized approaches. The key technique of filter-based localization approaches

is Bayesian filtering or its derivatives. Non-linear optimized approaches, mainly graph-optimization SLAM approaches, estimate the poses of the robot and landmarks of the environment to get the trajectory of the robot and the map of the environment. The cumulative error caused by various noises is eliminated by means of optimization.

On the other hand, the existing SLAM approaches can also be classified according to the sensor a robot carries. In fact, all of these approaches can be used to map the environment with specific premises considered. For instance, if a LiDAR sensor is only mounted on the robot, we can choose Cartographer, Gmapping, Karto_SLAM [31], and Hector_SLAM [32]. Furthermore, even for the same type of sensors, for example, if it is a single-line LiDAR, we can choose among Cartographer, Gmapping, Karto_SLAM, and Hector_SLAM. If it is a multi-line LiDAR, LOAM and its derivatives are a good choice. Considering the sensor that we currently mount on our robot, which is a single-line LiDAR, we aim to build a two-dimensional grid map. Thus, only Cartographer, Gmapping, Karto_SLAM, and Hector_SLAM were considered in the experiments.

Since Cartographer is well-known for its high mapping accuracy, we take Cartographer as a demonstration example for the proposed radiation-mapping method. The mapping process of the Cartographer can be divided into two parts: front end and back end, as shown in Figure 3. The sensor data are divided into LiDAR data, odometry data, and IMU data. The LiDAR data are preprocessed by a voxel filter to reduce the amount of LiDAR data, the IMU data are fused with the Odom data after the IMU tracker, and the fused data are transmitted to scan matching in the front end for positional interpolation. The front-end part accepts the processed sensor data and then performs the scan-matching process, which interpolates the current LiDAR data into the best position of the submap by matching the LiDAR data. The submap is continuously built with new LiDAR data being inserted. The front end treats the scan matching of local maps as a quadratic optimization problem, with the Ceres-solver solving the optimization problem. The back-end part contains the loop closure, which is implemented by using sparse pose adjustment (SPA) to eliminate matching errors between submaps. An important part of this process is the scan matching with the submap; this is performed by using the branch-and-bound scan matching (BBS) method, improving accuracy and speed considerably.

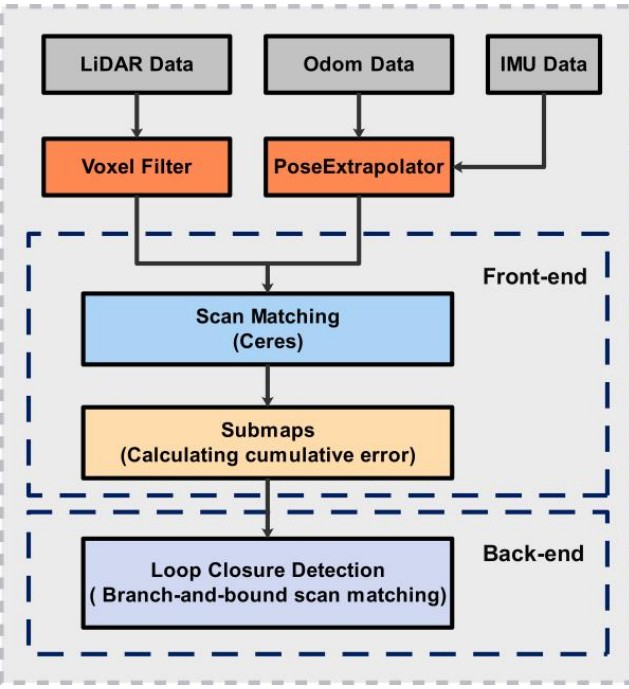

**Figure 3.** Flowchart of the Cartographer algorithm.

The main difference between Cartographer and other two-dimensional SLAM mapping algorithms is the use of submap in the scan-matching session. Whenever the scan data are obtained, they are matched with the most recently created submap, so that the LiDAR data of this scan can be inserted into the best position on the submap. The mapping process is run iteratively.

The poses of an autonomous mobile robot can be represented by $\xi = (\xi_x, \xi_y, \xi_\theta)$, with $\xi_x$ and $\xi_y$ representing translations in the $x$ and $y$ directions, respectively, and $\xi_\theta$ representing rotations in the two-dimensional plane. The scan data are added to the submap coordinate system through Equation (6):

$$T_\xi p = \begin{pmatrix} cos\xi_\theta & -sin\xi_\theta \\ sin\xi_\theta & cos\xi_\theta \end{pmatrix} p + \begin{pmatrix} \xi_x \\ \xi_y \end{pmatrix} \tag{6}$$

where $T_\xi$ represents the position pose transformation from the point cloud coordinate system to the corresponding submap coordinate system, which transforms the point $p$ in the point cloud from the point cloud coordinate system to the submap coordinate system.

Cartographer represents the submap by using a probability grid; each pixel stores the probability value of the occupied grid point. The occupancy probability values are updated according to the pose of each map point in the submap coordinate system. The Ceres-based scan matcher calculates the pose of the LiDAR-scanned point cloud in the submap, which converts the pose solution to a nonlinear least-squares problem, which is expressed as Equation (7):

$$\underset{\xi}{argmin} \sum_{k=1}^{K} \left(1 - M_{smooth}(T_\xi p)\right)^2 \tag{7}$$

where $M_{smooth}$ represents a bicubic interpolation function.

Cartographer achieves accurate mapping of large scenes by creating a large number of submaps. The pose estimates obtained by scan matching between the current laser scan and multiple neighboring laser scans are reliable in a short period but accumulate incorrectly over time. Consequently, loop closure is implemented to reduce the cumulative error. Cartographer applies the branch and bound optimization method to reduce computational cost and improve real-time performance. The optimization problem can be described as follows:

$$\underset{\xi \in \omega}{argmax} \sum_{k=1}^{K} M_{nearest}(T_\xi h_k) \tag{8}$$

where $\omega$ is a search window size, $K$ is the last scan point, $M_{nearest}$ is the extension of the nearest grid point in $M_{smooth}$, and $h_k$ represents the LiDAR measurement data.

*2.3. Methods*

Map fusion is the main contribution of this work, which focuses on a grid map and radiation map fusion. Figure 4 illustrates the map fusion process framework. After building the scene map using the Cartographer algorithm, the robot can locate itself in the scene. The mobile robot carries a surface contamination monitor to detect radiation source data on the scene. The position of the radiation source in the scene is determined by means of a transform frame (TF)coordinate transformation relationship, which enables the radiation source to be located. When the radiation data are transmitted to the robot operating system (ROS) via the data interface, the radiation data and grid map are fused by means of data-fusion methods to obtain detailed information on the radiation scenes, as well as the map information.

In order to realize the map fusion of the environment map and the radiation data, the data communication interface that can receive the radiation data collected by the radiation sensor needs to be designed. In this communication setup, the sensor communication protocol is transmission control protocol/internet protocol (TCP/IP). Message interoper-

ability is driven via a custom interface. Finally, when radiation information is detected, it is transmitted to ROS for algorithmic processing to complete the radiation-map building. Since different two-dimensional SLAM algorithms perform differently in their particular operating environment, we propose a framework for radiation mapping that is appropriate for the common two-dimensional SLAM. This framework allows the flexibility to switch algorithms depending on the environment in which two-dimensional SLAM is applicable.

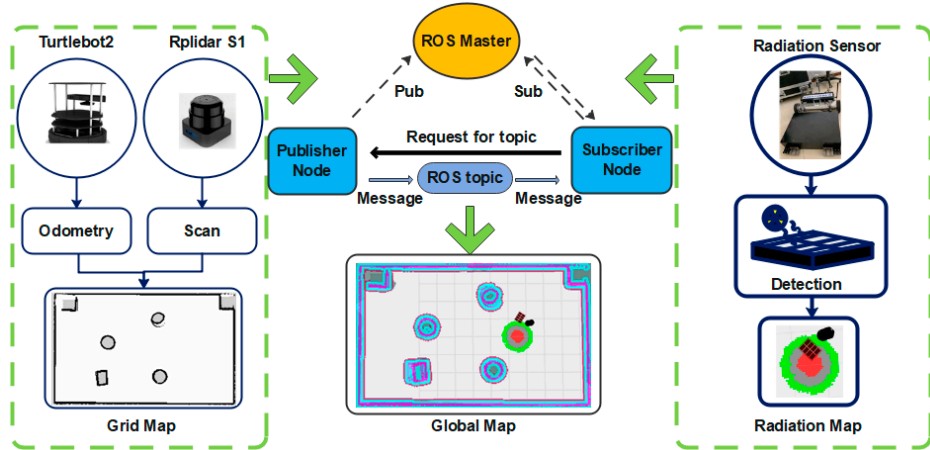

**Figure 4.** Map fusion framework.

The proposed map fusion approach takes reference from the Costmap approach [33], where the robot can obtain an all-around perception of the surrounding terrain's state through this map. Costmap is a type of map that is attached to the robot. It uses the robot as the coordinate origin at all times and represents the navigation environment around the robot through the probability values of the grid map.

As shown in Figure 5, where the static layer is the static map of the scene, the obstacles layer is the obstacles detected by the LiDAR during the detection process, the radiation layer is the radiation map layer detected by the surface contamination monitor, and the global map is a layer superimposed from the three mentioned layers. The radiation layer is initially a blank Costmap, which has the same dimensions as the grid map generated by Cartographer. Afterward, the radiation values are tagged into the radiation layer based on the radiation data returned by the interface callback during the robot's radiation detection. The radiation layers are superimposed on the global map for map fusion. During the detection process of the mobile robot, not only is radiation information added to the global map but also part of the obstacles is detected and updated to the obstacle layer. At the same time, the global map is updated with the obstacle layer.

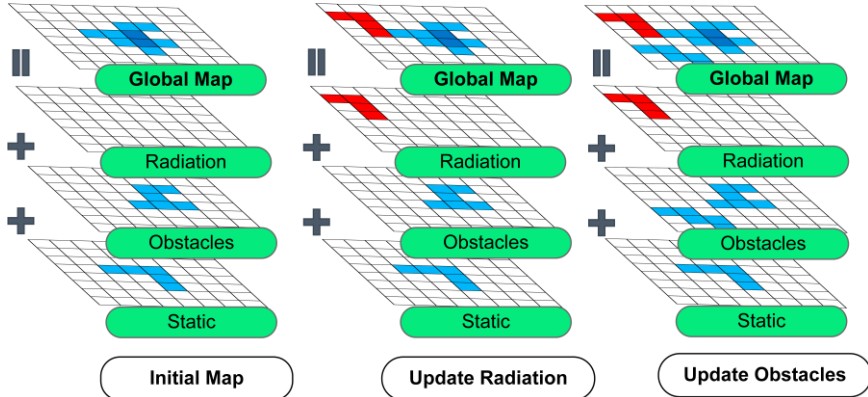

**Figure 5.** Radiation map's update process.

*2.4. System Modeling in ROS*

This part is about modeling the robot system in ROS. The first step is the model construction of the Turtlebot2 robot. The modeling results are shown in Figure 6. The Turtlebot2 was equipped with LiDAR for grid-map building. Furthermore, the model of the surface-contamination monitor was completed based on the characteristics of the sensor, and the feasibility of the model was verified in ROS.

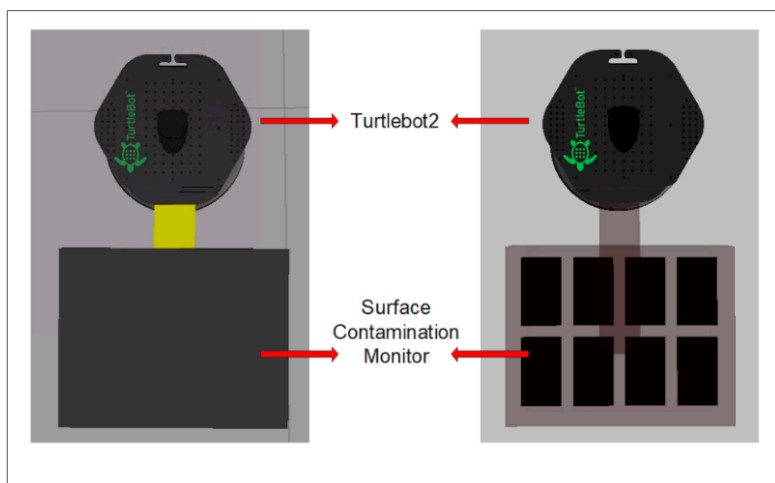

**Figure 6.** Modeling results for the robotics platform, with model construction results in Gazebo on the left and model visualization results in RVIZ on the right.

The model building process is shown in Figure 7. First of all, the physical parameters of the sensor are set and the sensor is modeled in Gazebo according to these parameters. Since the radiation sensor consists of eight detection windows, eight separate windows are added to the modeled robotics system. The communication interface described in Section 2.3 is set up for radiation data transmission.

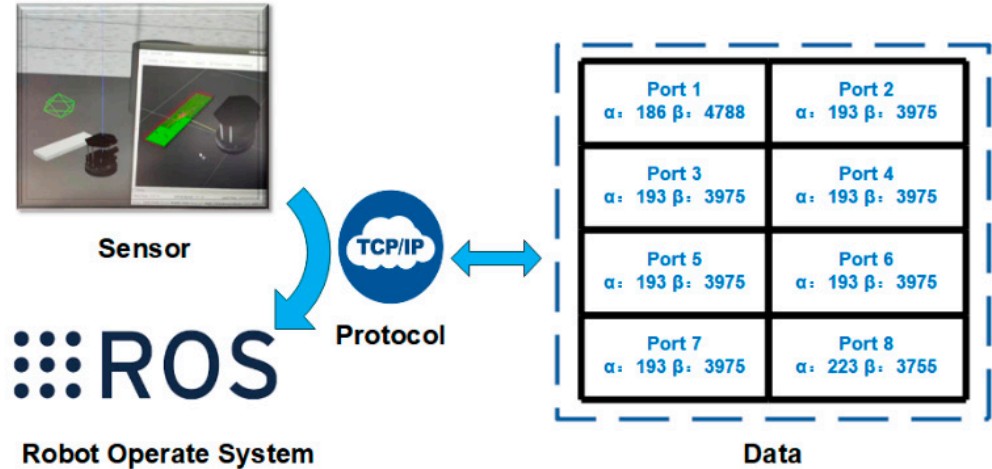

**Figure 7.** The process of communicating radiation-detection data.

**3. Experiments**

We conducted a group of experiments to verify the proposed method. Firstly, we tested four algorithms for two-dimensional SLAM on a public dataset and analyzed the performance of the four algorithms by using the EVO tool. Secondly, we completed the radiation mapping in a simulation environment, and the accuracy of the data detected by the radiation sensors was evaluated. Finally, we experimented with the abovementioned four two-dimensional SLAM algorithms in real-world scenarios and used Cartographer to build the radiation map in both indoor and outdoor environments.

### 3.1. Comparison of Different SLAM Methods on a Public Dataset

We evaluated the performance of the Cartographer by using the publicly available dataset MIT STATA [34], and its performance was compared with Gmapping, Karto_SLAM, and Hector_SLAM since they are all designed for two-dimensional mapping with single-line LiDAR.

This paper uses the EVO tool to analyze the accuracy of the map [35]. The EVO tool is often applied in SLAM to evaluate the performance and trajectory accuracy of systems. The absolute trajectory error is used to calculate the difference between the true value of the sensor pose and the estimated value of the SLAM system. The relative pose error is used to calculate the difference between the pose change within two time stamps. The mapping results of the four SLAM methods are shown in Figure 8. The comparison indicates that Cartographer builds the map with higher accuracy than Karto_SLAM and Hector_SLAM. Figure 9 demonstrates the absolute pose errors of the four SLAM methods in terms of RMSE, MEAN, MEDIAN, STD, MIN, MAX, and SSE, respectively. Table 3 shows the results of the absolute positional errors of the four SLAM methods.

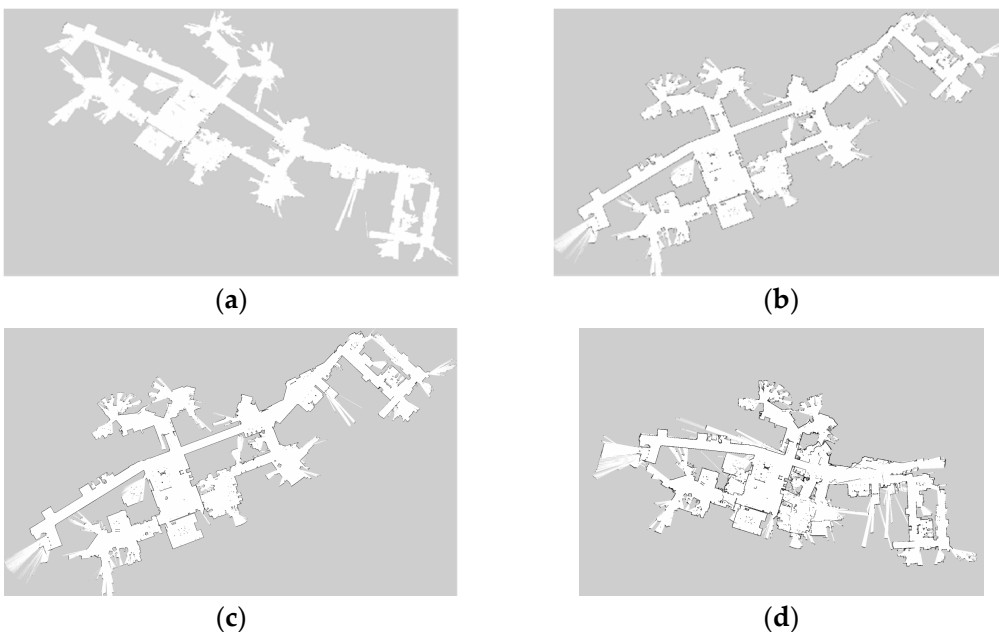

(a)  (b)

(c)  (d)

**Figure 8.** The results of the map building on the public dataset: (**a**) Cartographer, (**b**) Gmapping, (**c**) Karto_SLAM, and (**d**) Hector_SLAM.

**Table 3.** Statistical summary of absolute pose error.

|  | RMSE (m) | MEAN (m) | MEDIAN (m) | STD (m) | MIN (m) | MAX (m) | SSE (m) |
|---|---|---|---|---|---|---|---|
| Cartographer | 0.318046 | 0.244296 | 0.141310 | 0.20365 | 0 | 0.85211 | 2739.224 |
| Gmapping | 0.478861 | 0.444957 | 0.396801 | 0.176978 | 0 | 1.01641 | 6209.652 |
| Karto_SLAM | 0.816053 | 0.636724 | 0.427567 | 0.510436 | 0 | 1.65461 | 18188.1 |
| Hector_SLAM | 1.33959 | 1.14667 | 1.20988 | 0.692579 | 0 | 2.80609 | 49009.8 |

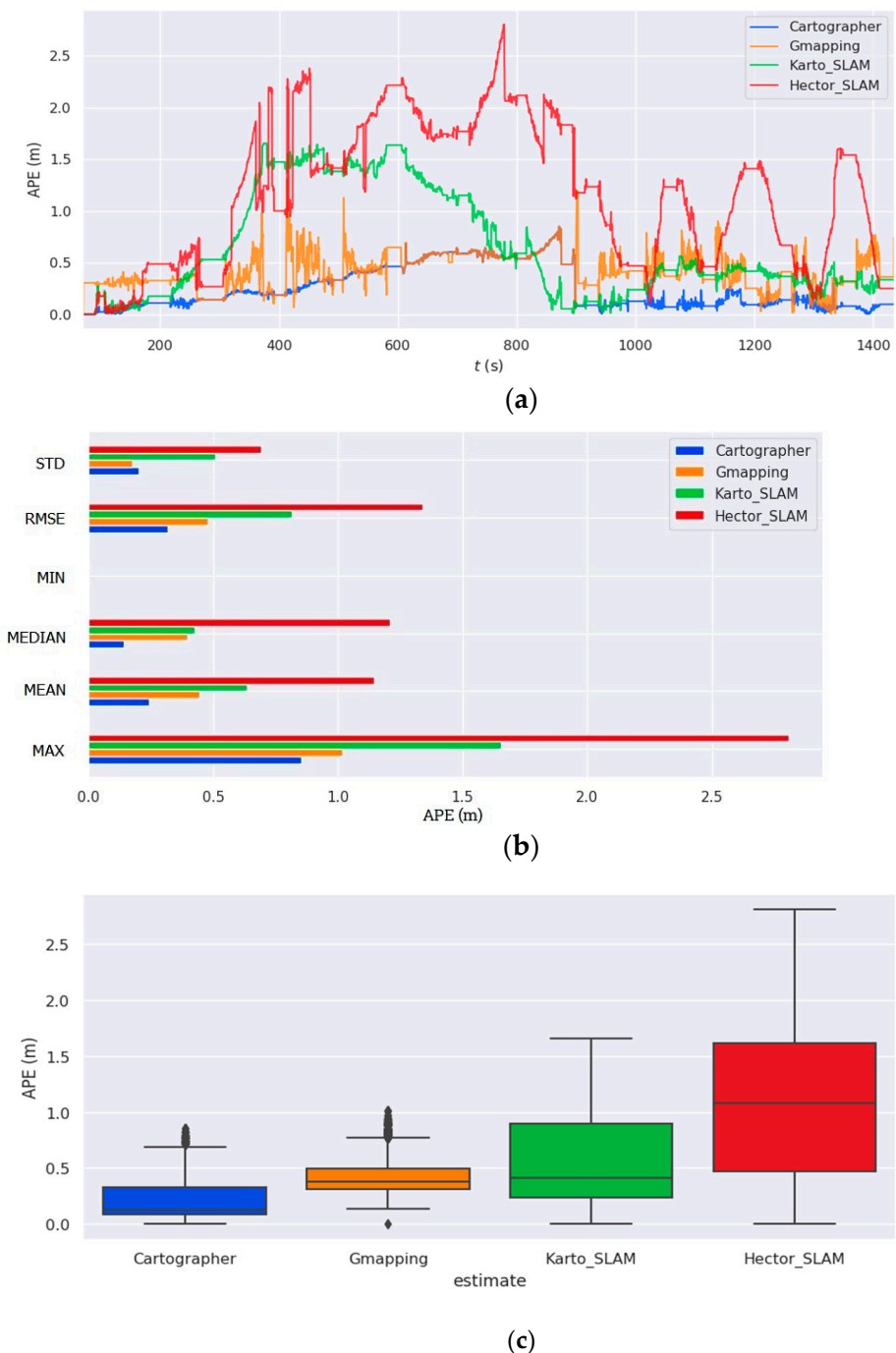

**Figure 9.** Comparison of absolute positional errors between multiple SLAM systems: (**a**) the graph of absolute pose errors of four SLAM methods, (**b**) statistic errors of four SLAM methods, and (**c**) box plots of four SLAM methods.

### 3.2. Radiation-Map-Fusion Results in the Simulation Environment

The results of the robot's map fusion during radiation detection are shown in Figure 10. The nuclear plant simulation environment used in Reference [36] was adopted, as shown in the Figure 10a,b. In this simulation environment, a single radiation source and multiple radiation sources are added to the simulation environment, respectively. The radiation-mapping results are shown in Figure 10c,d. Different radiation levels are set during the radiation-detection process, where green represents low-risk radiation, purple represents medium-

risk radiation, and red represents high-risk radiation. From the map-fusion results, we can see that the proposed radiation-mapping method shows promising results for α/β radiation.

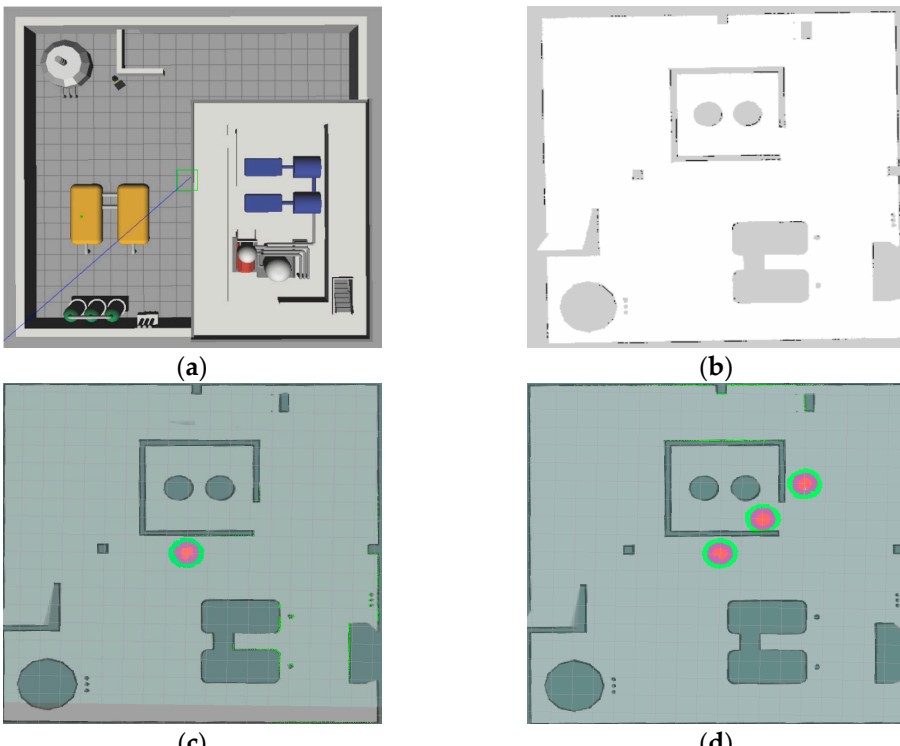

**Figure 10.** Map fusion results: (**a**) simulation environment, (**b**) grid map of the simulated environment, (**c**) single-radiation-source map-fusion results, and (**d**) multi-radiation-source map-fusion results.

Radiation levels are indicated according to different doses of incoming α/β radiation and are thus shown in different colors on the radiation map. Real-time monitoring of the radiation distribution can be achieved during the robotic detection of radiation, as shown in Figure 11. Figure 11a shows the real-time scenes captured by a camera during the detection process, and Figure 11b shows the real-time radiation-detection results. At the same time, real-time radiation detection allows the robot to be remotely controlled for radiation detection, thus protecting human beings from radiation or radioactive sources. As shown Figure 10b, we implemented a color-coded radiation-map-construction method during the robot's real-time radiation detection. We used different radiation cost values to distinguish radiation risk areas, where green represents low dose values, with a radiation interval of 5–2000 radiation dose values, and its radiation cost value is 102; purple represents medium dose values, with a radiation interval of 2000–4000 radiation dose values, and its radiation cost value is 58; and red represents high dose values, with a radiation interval of 4000–6000 radiation dose values, and its radiation cost value is 128. The differentiation of radiation intensity by color allows the riskiness of radiation areas to be effectively determined.

Figure 12 shows the error analysis of radiation data during radiation detection. This section shows the dose values for a single radiation source during the detection process in Figure 11b. Figure 12a shows a comparison between the measured radiation data and the true radiation data. The horizontal axis indicates the detection process over time, and the vertical axis shows the radiation dose values; the blue line is the true dose value, and the red is the actual radiation sensor detection value. Figure 12b shows the results of the radiation-detection error. The results implicate that the mobile robot detects radiation with high efficiency in the simulation environment, with a maximum error around ±0.7%, and the average error is 0.3%.

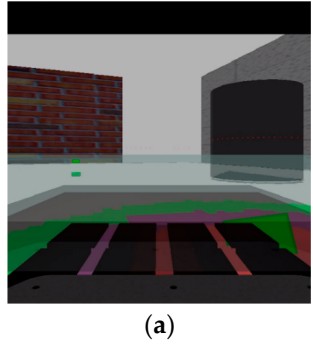 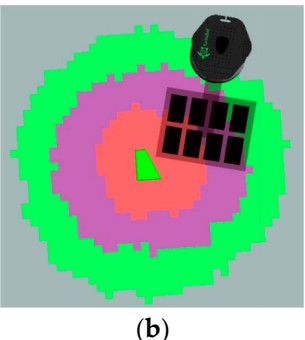

(**a**)           (**b**)

**Figure 11.** Real-time radiation-detection results in simulation: (**a**) camera view and (**b**) local radiation map.

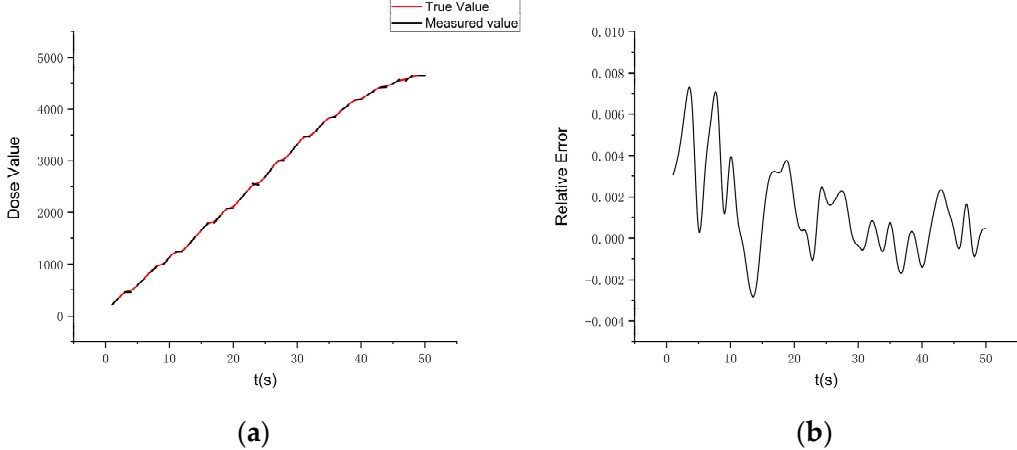

(**a**)           (**b**)

**Figure 12.** Analysis of radiation-detection data: (**a**) comparison of real radiation data with measured radiation data and (**b**) radiation-detection-data errors.

### 3.3. Radiation Map Fusion Results in the Real Environment

In addition to the simulation experiments, the radiation-mapping performance was also tested in the real environment. The test was conducted in both an indoor environment and an outdoor environment. The indoor environment is a long corridor in which Cartographer, Gmapping, Hector_SLAM, and Karto_SLAM were each tested separately. Figure 13a shows a scene from an indoor environment. Figure 13b–e show the map-construction results of the four SLAM methods, respectively. The fused mapping of the scene and radiation is shown in Figure 13f, where a single radiation source was set up for radiation mapping in an indoor environment, and the radiation-mapping result shows that the radiation source can be effectively located and detected. It is worth noting that the two-dimensional-grid map-building algorithm can be selected according to the size and complexity of the indoor environment, thus achieving higher building efficiency and accuracy in locating radiation sources.

The outdoor environment was a park located at Taiyuan University of Technology, as shown in Figure 14a. The scene-mapping results of Cartographer, Gmapping, Hector_SLAM, and Karto_SLAM are shown in Figure 14b–e, respectively. We can see that Hector_SLAM and Karto_SLAM build maps with a large drift in this environment. The reason is that these two SLAM methods have no advantage in larger outdoor scenarios. Figure 14f shows the results of the radiation mapping in the outdoor environment, where multiple radiation sources were set up to build a radiation map. We can see that the radiation sources can be located, and the radiation mapping result is displayed in different colors according to the level of radiation dose. With excellent performance in outdoor environments, Cartographer can accurately locate the radiation sources.

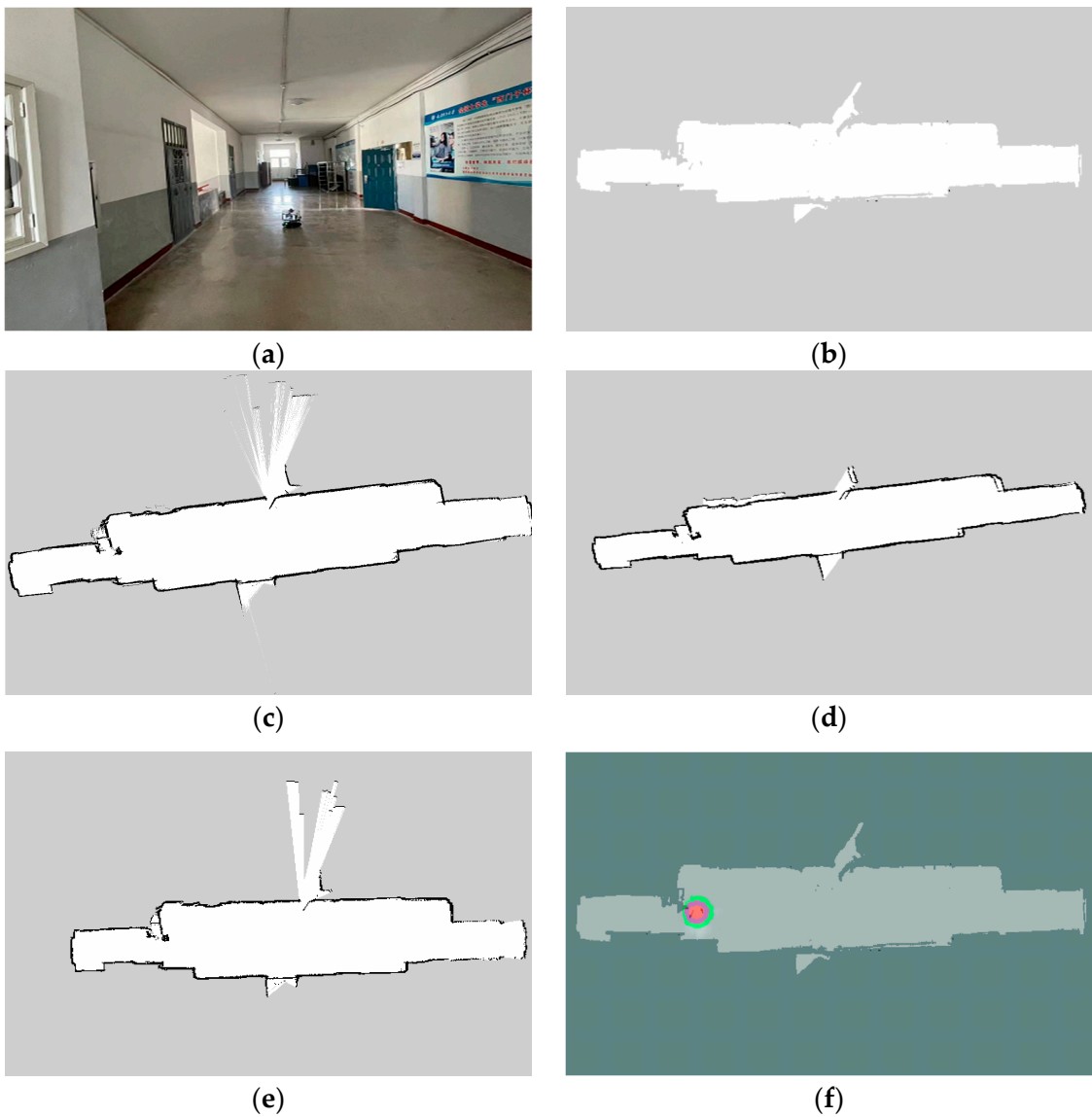

**Figure 13.** The result of the indoor scene map building: (**a**) the real environment, (**b**) the mapping result of Cartographer, (**c**) the mapping result of Gmapping, (**d**) the mapping result of Hector_SLAM, (**e**) the mapping result of the Karto_SLAM algorithm, and (**f**) the radiation mapping result of the indoor environment with one radiation source.

In the real environment, the mapping process is susceptible to a variety of errors. Therefore, choosing a suitable map-construction method can improve the localization accuracy of the radiation source. From the experimental results in Figures 13 and 14, we can see that Cartographer has a more balanced performance indoors and outdoors, and this is the reason for its selection in this paper. However, it is worth noting that the other three grid-mapping algorithms mentioned in this paper have advantages in some environments; for example, Gmapping shows the best mapping performance when it comes to small scenes. Therefore, the choice of method for two-dimensional grid-map building is flexible. The mapping results show that Cartographer has an advantage in building maps in both indoor and outdoor environments and that the method proposed in this paper has gained a satisfactory performance for radiation mapping.

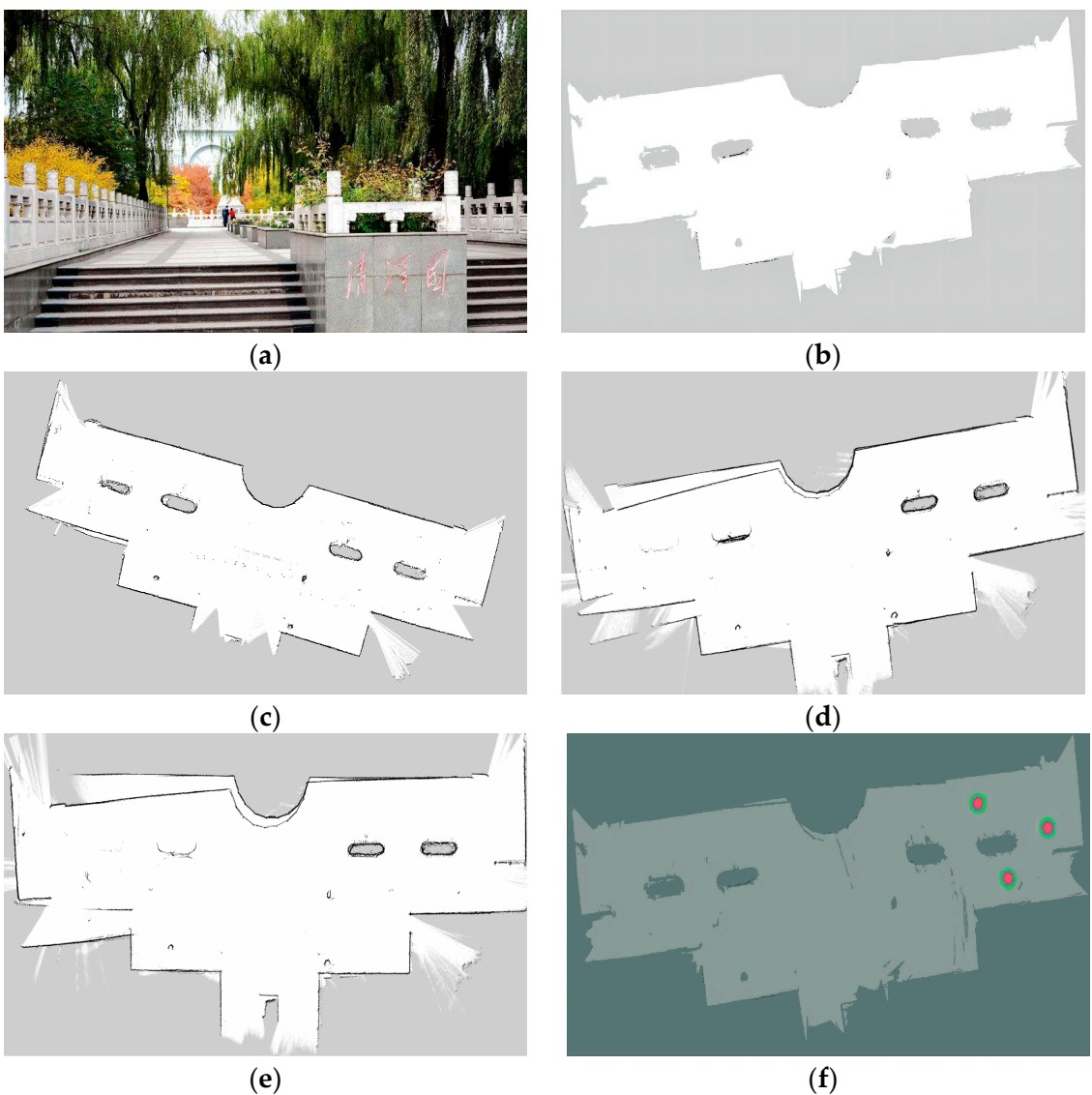

**Figure 14.** The result of the outdoor-scene map building: (**a**) the real environment, (**b**) the mapping result of Cartographer, (**c**) the mapping result of Gmapping, (**d**) the mapping result of Hector_SLAM, (**e**) the mapping result of the Karto_SLAM algorithm, and (**f**) the radiation mapping result of the outdoor environment with multi-radiation source.

## 4. Discussion

The α/β radiation-mapping method proposed in this paper is based on the integration of SLAM methods with nuclear-radiation-detection sensors, providing new means to map and visualize nuclear radiation for a friendly human–robot integration. It enhances the accuracy, speed, and efficiency of many applications related to nuclear- and radioactive-material detection and mapping; performs the monitoring of nuclear-legacy sites and operational nuclear facilities; offers wider radiation protection (e.g., accelerator facilities); provides consequence management for both intentional and unintentional releases of radioactive material; and leads to the decommissioning of nuclear facilities.

We proposed a general framework for autonomous radiation mapping through the integration of SLAM techniques and radiation-detection sensors, such as surface-contamination monitors. In this work, the two-dimensional grid map was built by SLAM methods in ROS. Four SLAM methods, namely Gmapping, Hector_SLAM, Karto_SLAM, and Cartographer, were specifically implemented to showcase the LiDAR-based SLAM methods for radiation mapping. From Figures 8 and 9, we can see that, in large-scale areas,

Cartographer is more accurate than the other three methods; this is due to the fact that Cartographer uses loop-closure detection for position correction during the mapping process. Among these four SLAM algorithms, both Gmapping and Cartographer have odometry. They use the odometer data to calibrate the map and for localization. Hector_SLAM requires a high level of sensor accuracy, which is the main reason for the large drift in the experimental results. Karto_SLAM has an intermediate performance on public datasets; however, real experiments have a higher time complexity compared to Cartographer.

For the radiation-detection sensors, we adopted the surface-contamination monitor developed by the China Institute for Radiation Protection in the robot for $\alpha/\beta$ radiation detection. Accordingly, we designed the data communication interface so that the radiation data can be collected by the radiation sensor and fused with the grid map built by SLAM methods. The radiation-mapping process can be described as follows. Firstly, the radiation sensors acquire radiation-information data and transfer the acquired radiation data to ROS via TCP/IP. Secondly, the data are parsed in ROS, and the processed data are passed to the radiation-map-construction algorithm for radiation-map construction. At the same time, the radiation map is merged into the two-dimensional grid map built by a SLAM method to locate the radiation source.

A color-coded radiation map was presented wherein different colors indicate different levels of radiation, allowing for a more intuitive display of the radiation map. We conducted the experiments in both the simulated environment and the real environment to demonstrate the radiation-mapping performance in both a single-radiation-source environment and a multiple-radiation-sources environment. At the same time, the error analysis of the radiation data during the radiation-detection process shown in Figure 11 indicates a promising detection accuracy, with a maximum dose-detection error of around $\pm0.7\%$ and an average dose-detection error of 0.3%.

In our current work, we demonstrated the feasibility of the proposed method for autonomous $\alpha/\beta$ radiation with SLAM techniques. Both the experiments in a simulation environment and actual environments show promising results for potential nuclear-radiation applications. However, more experiments need to be performed to test the performance in real nuclear-radiation scenarios; such experiments will be performed cooperatively with the China Institute for Radiation Protection in the near future. Moreover, we will work on expanding the framework for other types of radiation mapping, such as $\gamma$-rays.

### 5. Conclusions

This paper proposed a method that combines SLAM techniques with nuclear-radiation-detection sensors to address the drawbacks of traditional manual radiation detection. The main contribution of this paper is four-fold: (1) SLAM was adopted to achieve autonomous radiation detection; (2) a surface-contamination monitor developed by the China Institute for Radiation Protection was first adopted in a robot for $\alpha/\beta$ radiation detection; (3) a general radiation mapping framework was proposed for $\alpha/\beta$ radiation mapping, in which LiDAR-based SLAM methods were adopted to demonstrate the radiation-mapping performance; and (4) a color-coded radiation map was projected onto the environment map built by SLAM methods. The simulation and real-scene experiments verified the proposed method and showed a promising future in the applications of autonomous nuclear detection. In the future, more experiments need to be performed in real nuclear radiation scenarios so that the proposed method can be promoted for actual applications in nuclear plants or other places with potential nuclear leaks. Furthermore, our work will focus on extending this radiation-map-building method to highly radioactive particles, such as $X/\gamma$-rays, to obtain a flexible radiation-mapping platform for various nuclear radiations.

**Author Contributions:** Conceptualization, X.L., L.C. and Y.Y.; funding acquisition, L.C., X.X., G.Y. and Z.Z.; investigation, X.L. and L.C.; methodology, X.L.; project administration, L.C., X.X., G.Y. and Z.Z.; supervision, L.C.; validation, X.L.; writing—original draft, X.L.; writing—review and editing, L.C. All authors have read and agreed to the published version of the manuscript.

**Funding:** This research was funded by the National Natural Science Foundation China, grant numbers 62073232 and 61973226; Foundation for Scientific Cooperation and Exchanges of Shanxi Province, grand number 202104041101030; and the Natural Science Foundation of Shanxi Province, grant number 201901D211079.

**Institutional Review Board Statement:** Not applicable.

**Data Availability Statement:** Not applicable.

**Conflicts of Interest:** The authors declare no conflict of interest.

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
