# Peer review of "An Alpha/Beta Radiation Mapping Method Using Simultaneous Localization and Mapping for Nuclear Power Plants"

_machines, doi:10.3390/machines10090800_

Round 1

Reviewer 1 Report (New Reviewer)

The authors presented an autonomous α/β radiation mapping framework using a mobile robot carrying a LiDAR sensor and a nuclear radiation detection sensor. The proposed method was tested in both simulation and real environment in ROS for both indoor and outdoor environments. However, there are some suggestions about their work:

- Names of Subsections 2.2, 2.3 are not good enough for their content.

- In Subsection 2.1: Each part in Figure 1 has a name, so explanations in following paragraphs should include these names.

- In Subsection 2.2: Don’t need to present to much about SLAM concept (in this paper is LiDAR SLAM), because it is not the main contribution of this paper.

- In Subsection 3.1: Don’t need to spend too much time on this experiment, because it is not what readers desire to see in this paper.

- In Subsections 3.2, 3.3: should be investigated to enrich scientific contents.

- Should add one section to discuss in detail about your experimental results and contributions.

Good luck!

Author Response

Reviewer 2 Report (New Reviewer)

The publication is at a fairly high level. I have a few minor comments:

In my opinion

1. Figures 9 and 10 are difficult to read

2. It is worth emphasizing in the summary

3. The bibliography is often not up-to-date - too many items below 2010

4. The bibliography should be formatted in accordance with the journal's requirements

5. I understand that the authors intend to continue their research, but it is worth emphasizing in the summary how important they are and where in practice the method can be used, for what exactly

6. At the beginning of the work, I suggest adding the abbreviations used in the publication, it will enable the reader to understand the content of the publication more easily

In my opinion, the article is suitable for publication with minor changes

Author Response

Reviewer 3 Report (New Reviewer)

The authors propose a combination of SLAM and a radiation detection method. Thanks to SLAM an autonomous character of detection can be achieved. The author's intention is to apply the proposed method to robots for α/β radiation detection. Usability of the method was simulated and verified experimentally. The method is said to have a potential to be extended to other kind of radiation (X-rays/γ-rays).

There are several comments related to the reviewed manuscript:

All physical quantities used in equations (1) and (2) have their physical units specified except for the surface emissivity response (Ri, Line 171). For completeness, it would be good to add it as well.

To increase the paper quality, I recommend performing a general language revision, removing typos and similar errors. To demonstrate and support this position, I present recommendations for the first 100 lines below:

1: … a focal point in the field …

 16: … radiation detection sensor ...

23: … used in the new energy generation field.

31: … α-rays, β-rays, protons, and uncharged particles…

35: … to the human body.

39: … indirect detection, and scan …

42: … On the other hand, the scan radiation….

47: … results on an environment map…

52: … unmanned systems since it does…

55: … detector, and build a map …

60-61: … as sparse feature maps, point cloud maps, topological maps, and grid maps … Among these, a grid map is…

63: … as well as follow a…

65: … Moraves [10].

66: The idea is to divide… and propose…

68: Konolige [11] and Thrun [12]…

70: .. Grisetti [13], …

72: The Gmapping algorithm…

73: … open-source SLAM algorithm is …

79: … in a grid map to…

81: … environment-aware sensor fusion [17]. …

88: … source in a three-dimensional…

94: … methods, and Bayesian…

95: … to X-rays, and γ-rays… to a much further distance compared to …

Round 2

Reviewer 1 Report (New Reviewer)

The authors have spent much time to improve the quality of this paper, in terms of conciseness and scientific enhancement. There are some suggestions about their work:

- In Table 1: Abbreviations should be written as capital letters, e.g. RMSE, STD, ...

- Remember to define all abbreviations before using them, e.g. TF, APE, ...

- Figure 3: I don’t know why you mentioned Odom data (Odometer data) and IMU (IMU data) in a figure of LiDAR SLAM subsection. Usually, a figure of a SLAM system has two parts: front-end for processing data from SLAM sensor and back-end for optimization (including loop closing).

- Figure 3: Correct them: LiDAR, IMU data, typing error in the last block.

- Table 3: Write down their units.

- Names of Figure 9 and Table 3 seem similar. I suggest that Figure 9a is “graph of”; and Table 3 is “statistical summary of”.

- Line 402: Should be Figures (13) – (14).

- In Subsection 3.3: Please add some evaluations about results in Figures 13f and 14f.

Author Response

This manuscript is a resubmission of an earlier submission. The following is a list of the peer review reports and author responses from that submission.

Round 1

Reviewer 1 Report

The authors propose an information fusion approach to achieve SLAM that includes alpha/beta radiation in the map. The base framework is the Gmapping algorithm for SLAM. Information from a radiation sensor is incorporated in the occupancy grid, so that a radiation map can be constructed.

While the authors describe a system that can be used in actual experiments, all testing is performed only in simulation. It is unclear if the radiation sensor is accurate or reliable. Without experiments, it is not possible to measure how successful this work is. Besides, the radiation data is used as an additional, uncoupled layer of the map, so the base SLAM algorithm is not altered in any way. 

The authors should attempt to incorporate the radiation data into the SLAM algorithm, and conduct actual experiments with small, controlled radiation sources. 

Author Response

Thanks for the reviewer’s constructive suggestion. As α/β radiation dies out when it’s even a few centimeters away from the radiation source, we take the position where the radiation is detected as the location of the radiation source. In this paper, we try to demonstrate that SLAM can be considered as a method to realize autonomous nuclear detection.

The real environment experiments were conducted for the performance analysis of SLAM methods in the revised version. Unfortunately, the actual experiments with real controlled radiation sources were not able to conduct currently since the Surface Contamination Monitor we used was not lent to another group for a different research purpose. We will and have to conduct this experiment once the Monitor is available since this is a real application-oriented research.  

Reviewer 2 Report

Thanks for sharing your work. Please find my comments attached in the pdf.

Author Response

We thank the reviewer for his/her precious time in reviewing our manuscript and for providing instructive comments and suggestions. We have updated the manuscript according to the review comments /suggestions. The point-by-point responses to the comments/suggestions are listed in the pdf. We hope the reviewer will be satisfied with the revised manuscript

Reviewer 3 Report

The paper evaluates SLAM technology and presents a method for radiation mapping. The authors used existing SLAM algorithm (Gmapping). The contribution of the paper is the special application area and the fusion of "grid map' with radiation map, however the latter is only a spatial transformation of the data ("TF coordinate transformation" is a library specific term, a regular reader won't understand it) as I understood from the paper. This transformed map is then fed back to the SLAM algorithm. This hardly could be called a contribution. The algorithm is only tested in a simulated environment. The overall contribution of the paper to the field is low in my view. Nevertheless, given that the journal is open access and the paper is well-written and does not contain major flaws, I recommend it for publication.

Author Response

Thanks for your comments. We added experiments in real environments in the revised version, more details see p14, section 3.2.2. Experiments for practical use will be conducted in the near future.

Reviewer 4 Report

Interesting approach in its context of application for radiation detection. Nonetheless, there many aspects which does not contribute sufficiently to any novel point in the techniques; there are actually very mainstream and somehow old (SLAM framework). Finally, the few results presented are simulated. The work itself reduces to a fusion of a radiation sensor into Gazebo. There are many open lines for authors to work in the near future to enhance the quality of the paper:

Authors should investigate in depth, at least to compare benefits of other SLAM implementations. Gmapping is known to be valid but not enough in certain context.

First, how about its performance in a real environment? This should be tested.

Second, indoors/outdoors scenarios and comparison.

Third, there are more up-to-date approaches based on Lidar and graph SLAM such as GT-SAM, LISAM, LiODOM, and many others with support to vision (ORB2, etc), which should be compared, even under publicly available datasets (radiation data can always be simulated).

Materials and Methods: SLAM is a mainstream which does not require to be detailed in depth eq by eq

Experiments: radiation measurements have to be better explained. How is the real data (true values) estimated? Is also simulated as the robot scenario? Or are real results obtain in the same simulated scenario, which is actually the real one. These questions arise after observing Sec 3.2.2, so a better explanation should be made.

Author Response

Response: We are very grateful to the reviewer for his/her precious time in reviewing our manuscript and for providing valuable and positive comments. We have updated the manuscript according to the review comments /suggestions. The point-by-point responses to the comments/suggestions are listed below. We hope the reviewer will be satisfied with the revised manuscript.  

Comment 1: Authors should investigate in depth, at least to compare benefits of other SLAM implementations. Gmapping is known to be valid but not enough in certain context.

  • First, how about its performance in a real environment? This should be tested.
  • Second, indoors/outdoors scenarios and comparison.
  • Third, there are more up-to-date approaches based on Lidar and graph SLAM such as GT-SAM, LISAM, LiODOM, and many others with support to vision (ORB2, etc), which should be compared, even under publicly available datasets (radiation data can always be simulated).

Response: Gmapping, GT-SAM, LiODOM, LOAM, Cartographer, ORB-SLAM series, RTAB-MAP, Hector_slam etc. are compared in the revised version, see p5 to p6, from line 198 to line 223. We also added comparison on public Datasets among Gmapping, Hector_slam and Karto_slam, see p10, Figure 10.

We noticed there are many different SLAM approaches according to the different sensors a robot carries. Actually, all these approaches can be used to map environment with specific premises considered. For instance, if a LIDAR sensor is only mounted on the robot, we can choose Gmapping, Karto_slam, Hector_slam. If a camera is considered as the only sensor in a robot, we can choose ORB-SLAM series. For the same reason, if multi-type sensors are used in a robot, for example, IMU, camera and LIDAR etc, other SLAM approaches such as Vins-Mono, RTAB-MAP will be better choices. Furthermore, even for the same type of sensors, for example, LIDAR sensors, if it is a single thread LIDAR, we can choose among Gmapping, Karto-SLAM, and Hector-SLAM. If it is a multi-lines LIDAR LOAM and its derivatives will be a good choice. Considering the sensor we currently mount on our robot, which is a single line LIDAR, we aim to build a 2D-grid map. Thus, only Gmapping, Karto_slam, and Hector_slam are considered in experiments.

  • The performance of Gmapping in a real environmentis tested. We tested both indoor and outdoor real environments.The result is shown in Figure 16 and Figure 17. The analysis in given in p14, from line 386 to line 402.
  • The performance of Gmapping in indoors scenarios are tested in SLAM Benchmarkings datasets.The result is shown in Figure 10. The analysis in given in p9, from line 307 to line 319.
  • In this paper, considering the LIDAR we use in our Turbot2 is only a single line LIDAR for 2D mapping and the reason that we are still on the early stage to show readers that SLAM methods can be used to realize autonomous nuclear radiation detection, we take Gmapping as only an example for its readily implementation in ROS. In further study and practical use in real environment, we’ll consider other more suitable SLAM approaches according to the potential clients’requirement on sensors and performance.

Comment 2: 2.Materials and Methods: SLAM is a mainstream which does not require to be detailed in depth eq by eq.

Response: This part has been rewritten, see p5, from line 191 to line 197. However we still keep the part of explaining the main idea of SLAM for the convenience of researchers in the field of nuclear radiation detection. 

Comment 3: Experiments: radiation measurements have to be better explained. How is the real data (true values) estimated? Is also simulated as the robot scenario? Or are real results obtain in the same simulated scenario, which is actually the real one. These questions arise after observing Sec 3.3.2, so a better explanation should be made.

Response: The radiation measurements in the paper are simulated according to equation (1-4). In real scenario, the radiation data is similar to the simulated data but more complicated. In practical use, we can always use the radiation detector, which has been specified by China Institute for Radiation Protection to measure the radiation dose.

Round 2

Reviewer 1 Report

From the authors' response from my first round of comments, it is clear that no further experiments can be performed. This work does not indicate that the radiation data is influencing the SLAM algorithm in any way. Hence, this work is an application of SLAM algorithms to a particular problem but there is nothing specific to radiation detection in the algorithm, since the detection data is simply added as an extra layer to the occupancy map generated by any of the SLAM techniques shown in the work. As a result, there is no clear contribution.

Reviewer 2 Report

Thanks for responding to my comments point-wise and making amendments to the paper. Apologies, for the comment regarding Figure 13, I meant providing a colorbar for the dose; similar to the one already provided in Figure 12. For instance, how would one figure out which color has approximate what value of dose in Fig. 13(b)?

Thanks!

Reviewer 4 Report

This paper remains the same at this essence. Comparison is provided, but by simply using a benchmark tool provided by ROS where standard errors and performance of different algorithms is presented. However, there's no real implication on how the comparison applies over the performance of such SLAM method fused together with their approach.

This sort of comparison is more than mainstream in the field. It is needed that their own implementation is compared under different frameworks. So far this is just a well known SLAM approached which receives as input data, somehow of simulated data (tagged as "radiation").

There's very few novelty on that.

There are some mistakes like Fig. 14.